# Corrosion Modeling of Magnesium and Its Alloys for Biomedical Applications: Review

**Moataz Abdalla [1], Alexander Joplin [1], Mohammad Elahinia [2] and Hamdy Ibrahim [1,*]**

[1]  Mechanical Engineering, University of Tennessee at Chattanooga, 615 McCallie Ave,
    Chattanooga, TN 37403, USA; fjx788@mocs.utc.edu (M.A.); wcf761@mocs.utc.edu (A.J.)
[2]  Mechanical Industrial and Manufacturing Engineering, University of Toledo, 2801 W. Bancroft St. MS 312,
    North Engineering 2045, Toledo, OH 43606, USA; mohammad.elahinia@utoledo.edu
*   Correspondence: hamdy-ibrahim@utc.edu

**Abstract:** Biodegradable metals have been under significant research as promising alternatives to the currently in-use nonbiodegradable materials in the field of supportive medical implants. In this scope, magnesium and its alloys were widely investigated due to their superior biocompatibility over other metals. Most of the research effort in the literature has been focused on assuring the biocompatibility, improving mechanical properties, and tailoring the corrosion rate of magnesium-based implants. Furthermore, considerable research was done to develop numerical models towards an inexpensive and fast designing tools capable of simulating the degradation/corrosion behavior of magnesium-based implants. Due to the complexity of the degradation process and the various factors that can be involved, several hypotheses were introduced to provide a realistic simulation of the corrosion behavior in vitro and in vivo. A review of the current literature hypothesis and different modeling constitutive equations for modeling the corrosion of magnesium alloys along with a summary of the supplementary experimental methods is provided in this paper.

**Keywords:** biodegradable; magnesium; bioresorbable; modeling; corrosion; implants

## 1. Introduction

Biodegradable materials have been investigated recently for various applications due to their eco-friendly characteristics [1,2]. Metals such as magnesium, iron, and zinc are a branch of biodegradable materials class that also include biodegradable polymers and ceramics [3]. These biodegradable metals represent an important part of almost all the living organisms' dietary systems [4]. In the case of supportive implants, biodegradable metals are expected to provide the needed support for damaged tissue during recovery, then subsequently degrade and ideally are expected to be absorbed gradually and safely in vivo after healing [5]. For instance, bone fixation hardware (e.g., screws, nails, wires, and plates) is currently made of stiff and non-degradable metals such as stainless steel, titanium alloys [6], and Ni–Ti alloys [7]. These implants are essential to hold opposing segments of fractured bone still during the healing period by providing internal or external support to the fractured bones [8,9]. The high stiffness of standard-of-care fixation hardware, which is 5–11 times stiffer than the bone, may subsequently result in a failure to establish normal loading patterns [10,11]. Such abnormal loading patterns may result in one or more of these poor outcomes: (i) stress shielding and resorption of newly healed bone, (ii) stress concentration in the fixation device and device failure (e.g., plate cracking or screw pull-out), and/or (iii) inflammation and post-surgery infection. In some cases, bone implants can be left inside the patient's body without causing complications. However, in most other cases and due to the prementioned problems, bone implants are usually removed with subsequent surgeries. It is worth mentioning that biodegradable implants can be only used for regenerative surgeries that

require temporary support during recovery such as vascular stenting, bone fixations, and bone grafting. Non-degradable (permanent) implants are the only options in cases when a complete replacement of the tissue is required such as teeth and joints replacement [12].

An ideal biodegradable implant is expected to have a matching stiffness to that for bone then be absorbed entirely and safely after fulfilling the job of supporting the healing tissue leaving no residues; hence, no subsequent surgeries are required [12]. It is noteworthy that there is increased research activity in this area. For instance, the International Symposium on Degradable Metals has been held every year since 2009. In the United States, one-half of all the disease or injury-related hospitalizations and 72% of musculoskeletal injury charges during 2012 were accounted for orthopedic trauma with more than $214 billion in total hospital charges in 2012–2014 [13]. The trauma device market including general internal and external fixators was valued around $6.9 billion in 2017 in the United States [14]. One of the most promising materials for bone fixation applications is magnesium and its alloys. Magnesium has been a focus of interest for its medical use as early as 1878 [15]. Magnesium is currently the most investigated material due to its superior biocompatibility and favorable mechanical properties compared to other biodegradable polymers and metals such as zinc and iron. For example, the recommended dietary allowance (RDA) of magnesium for an average adult is 6 mg/kg per day [16], which is about 400 mg/day for adult males, while zinc RDA is 11 mg/day and iron RDA is 8 mg/day [17]. The relatively high intake of magnesium reflects its superior biocompatibility. Despite its advantageous properties, magnesium is one of the most active metals making its degradation rate very high in aqueous solutions like body fluids compared to other biodegradable metals (i.e., iron and zinc). Extensive research has been done on different techniques to address this issue such as alloying and coating [6].

In order to investigate the modifications applied to the magnesium to qualify it for bone fixation hardware, a lot of time-consuming and costly experimental work is needed to verify its degradation rate and mechanical integrity. To address this issue, several research groups have been working on developing numerical methods capable of simulating the degradation behavior of magnesium and its alloys in vivo. Proposed numerical models in literature can be categorized by the modeling method as (i) phenomenological methods and (ii) physical methods. Each method has its constitutive equations that are usually solved using the finite element method. Models domains varied between 2D and 3D. Models must be calibrated vs. experimental data and preferably verified using different geometries.

In this paper, a review of the current literature describing the nature of the degradation mechanisms of magnesium is presented. Then, a brief introduction to the currently available mechanisms is provided and followed by reviewing the various modeling techniques that are related to these mechanisms. In addition to the physical and phenomenological models, the cellular automata modeling method is also introduced as a potential method for biodegradation modeling. A review on the application of the level set method in this area of research is provided. Finally, a summary of all the experimental methods as in vitro and in vivo to calibrate and validate these models is covered.

## 2. Corrosion of Magnesium and Its Alloys

Magnesium is one of the most active metals and the least noble in the galvanic series with a standard electrode potential of −2.37 V [18]. This makes magnesium and its alloys very susceptible to galvanic corrosion. Galvanic corrosion is usually activated due to the presence of impurities or agglomerated cathodic secondary phases in the microstructure. It can also be activated when the material gets in contact with a nobler material inside a conductive media, which leads to local corrosion around the contact area. The effect of secondary phases was found to be minimized by heat treatment that turns the lamellar and spherical secondary phases into finely dispersed precipitates [19,20]. In addition, the addition of alloying elements such as Mn and Zr was found to improve the corrosion resistance of magnesium alloys (e.g., Mg–Zn-based alloys) by dissolving the insoluble impurities such as Fe and Ni into less active phases [6]. It was also established in the literature that grain refinement is helpful in enhancing corrosion resistance by alloying or mechanical treatments [6,21].

Corrosion resistance was found to be less in aqueous environments such as body fluids compared to atmospheric environment [22]. This is attributed to the development of quasi-passive magnesium hydroxide byproducts on the surface and hence reducing the corrosion rate. The formed magnesium hydroxide can react with chloride ions to produce highly dissolvable magnesium chloride. On the other hand, magnesium can develop a protective passive oxide layer in an atmospheric environment that can protect some magnesium alloys even in the marine environment [22]. High purity alloys such as AZ91E were found to provide corrosion resistance up to 100 times the corrosion resistance of a regular quality alloy [22]. Many different coatings were discussed in this review paper [6] to improve the corrosion resistance of magnesium and its alloys and to improve its biocompatibility towards biodegradable implants. So to summarize, corrosion of magnesium and its alloys is mainly function in (i) the level of harmful impurities namely, Fe, Ni, and Cu; (ii) composition, size and distribution of secondary phases; (iii) grain size; (iv) environment; and (v) the type and thickness of the coating. In order to develop models of the corrosion of magnesium, a comprehensive review on the current understanding of corrosion mechanisms of magnesium is provided in this section.

### 2.1. Galvanic Corrosion

Galvanic corrosion occurs in the least noble metal when two metals of different electrochemical voltage become in contact while inside a conductive medium. Since magnesium is the least noble metal, it is always consumed by anodizing. Electrons migrate from magnesium alpha phase to different cathodes releasing magnesium ions that diffuse to the surface and form corrosion products or dissolve in the surrounding medium in the case of the aqueous environment [22]. Magnesium can also be subjected to internal galvanic corrosion as well as external galvanic corrosion [22]. Internal galvanic corrosion or micro-galvanic corrosion is due to the presence of grains of impurities and cathodic phases on the grain boundaries as shown in Figure 1a. External galvanic corrosion occurs due to the contact with a nobler metal as shown in Figure 1b. α grains are either pure magnesium grains or solid solution of the magnesium and alloying elements such as Al, Zn, Ca, Mn, and some rare earth elements [23]. For a list of alloying elements and their effect on the corrosion properties, refer to this review paper [22]. β phases are the secondary phases that form on the grain boundaries and between grains. For example, $Mg_{17}Al_{12}$ is the β phase in the case of Mg–Al-based alloys such as AZ91 alloy. $Mg_{17}Al_{12}$ was found to affect the galvanic corrosion in two opposite ways. Song et al. [22] found that the anodic current of this phase is much less than that for the α phase, which suggests its relative passivity. However, it acts as the cathode that accelerates the corrosion rate of the alloy. Hence, they suggested that a large volume fraction of this β phase near the exterior might enhance the corrosion resistance while a small fraction can reduce it.

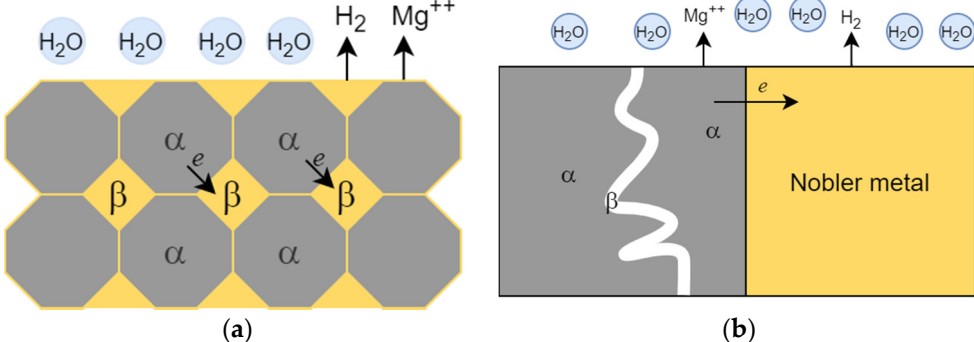

**(a)** **(b)**

**Figure 1.** (**a**) Schematic of internal micro-galvanic corrosion. (**b**) External galvanic corrosion.

### 2.2. Pitting Corrosion

Pitting corrosion is the second most common corrosion type of magnesium and its alloys. It is defined as a localized and random severe corrosion on the surface of the magnesium alloy (e.g.,

Mg–Al-based alloys) [22]. It is observed as shallow pits on the metal surface and follows after breakage of the protective passive layer on the surface after immersion in an aqueous environment with corrosive ions such as chlorides, see Figure 2. Unlike pure magnesium, magnesium alloys develop micro-galvanic cells around impurities or secondary phases near uncovered areas. Consequently, a spread of corrosion along the magnesium matrix surrounding particles or grains near the surface results in a cut out and dissolution of large particles of the metal into the environment. Corrosion proceeds widening the pit volume until a new passive corrosion product such as Mg(OH)$_2$ is precipitated on the surface [18], see Figure 2.

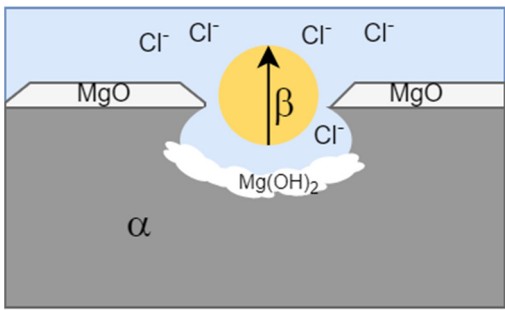

**Figure 2.** Schematic of pitting corrosion.

### 2.3. Stress Corrosion Cracking

High purity magnesium [24] and magnesium alloys [25] were found to be susceptible to stress corrosion cracking (SCC) that mainly depends on the applied stress level, the alloy composition, and the environment of operation. A good review of the types and reported factors that affect the SCC of magnesium is introduced here by Winzer et al. [25]. In order for SCC to occur, a threshold value of stress shall be applied depending on the alloy type. Some of the reported experimental values are summarized in this review [25]. SCC in magnesium has two categories: (i) intragranular stress corrosion cracking (IGSCC) due to continuous dissolution, and (ii) transgranular stress corrosion cracking (TGSCC) due to sequential discontinuous cleavage fractures [25].

IGSCC due to dissolution in Figure 3a, is hypothesized in three subcategories:

(i) Preferential attack (i.e., near the surface), in which the matrix is preferentially attacked and the adjacent grain boundaries to cathodic phases corrode creating small cracks near the surface. Applied stress opens the cracks and allows species in the solution to flow towards the crack tip which accelerates the crack growth due to the galvanic corrosion [25].

(ii) Galvanic corrosion due to passive film rupture, in which strains cause rupture of the protective oxide film and expose parts of the anode matrix. This creates a galvanic cell with covered cathodic regions which in return dissolves the matrix grains and initiates a crack through the grains.

(iii) IGSCC is initiated due to tunneling, which is a tubular pitting corrosion. These tunnels can be near each other leading to a ductile tear of the metal in between due to stress, which initiates cracks on the surface. Opened cracks will continue growing under cyclic loading and are also subjected to the formation of new pits.

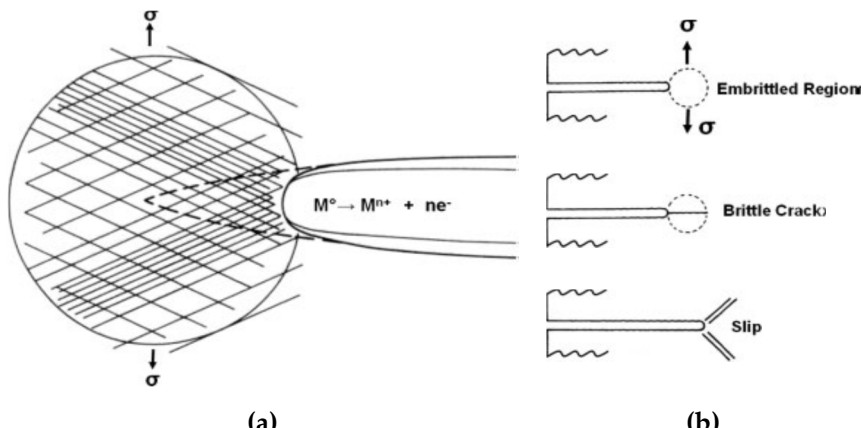

**(a)**                                                                                  **(b)**

**Figure 3.** (**a**) Stress corrosion cracking (SCC) due to dissolution. (**b**) SCC due to cleavage cracks, reprinted with permission from [25].

TGSCC as shown in Figure 3b, is more observed and has two common modes [25].

(i)     The first is cleavage fracture due to stages of electrochemical and mechanical effects. Electrochemical effects cause the initiation of pits that is followed by a mechanical effect in high-stress concentration that starts a cleavage crack. The crack propagates through the grain until it reaches an obstruction such as the grain boundary. Pitting corrosion then continues to initiate another crack in a different direction.

(ii)    The second main mode is hydrogen embrittlement. The evolution of hydrogen due to the electrochemical reaction (12) during the galvanic corrosion leads to embrittlement of a crack tip and propagation of cracks. Another hypothesis is that hydrogen reacts with magnesium resulting in a brittle magnesium hydride.

## 3. Modeling Methods

In order to model the different corrosion mechanisms, there are two main approaches in the literature. The first approach is considered phenomenological since it models the external visual corrosion effects on the metal such as pits, cracks, and general uniform corrosion. This approach is based on the continuum damage theory. The second is considered physical since it focuses on the physics of the species interaction and the governing electrochemical relations. Physical modeling can be used to study the uniform corrosion of the whole implant or as localized in small samples to study the pit growths [26].

### 3.1. Phenomenological Modeling

This approach is based on the continuum damage theory. Lemaitre et al. [27] introduced the damage concept in order to calculate the effective stress in a loaded structure with internal geometrical discontinuities. Those defects are in the shape of micro-cracks that are generated due to the accumulation of dislocations on one point causing debonding between the grains which on macro-scale reduces the bearing area of the material. To this end, they introduced a new scalar field $D$ that ranges from 0 to 1 to describe the status of the internal geometry over time with damage evolution formulae. Zero is for an undamaged material element and one for a completely damaged element, to compensate for the losses in the area, and to calculate the effective stress as in Equation (1) [27]. Shown in Figure 4, the distribution of this factor in the model by Amerinatanzi et al. [28] and the feature of hiding damaged elements. Since the highest stress that the material can take under tension is equivalent to its ultimate

strength value, the maximum value of $D$ (i.e., $D_{\text{critical}}$) in Equation (1) is obtained by substituting the effective stress by the material ultimate strength and solving for $D$ which is less than 1.

$$\widetilde{\sigma} = \frac{\sigma}{1 - D} \tag{1}$$

where $\sigma$ is the true stress and $\widetilde{\sigma}$ is the effective stress.

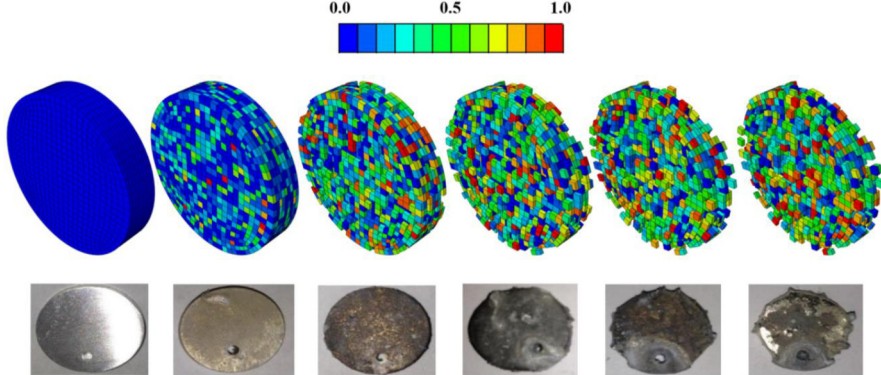

**Figure 4.** A qualitative comparison between the CD model results and samples subjected to immersion test, reprinted with permission from [28].

By applying this concept to account for mass loss over time due to corrosion using the finite element method, research groups could model the damaged elements in the mesh by introducing damage evolution formulae based on some material characteristics and the type of corrosion being modeled. The basic idea in order to model the corrosion of material is to calculate the scalar field $D$ for each element on the surface of the finite element model at each time increment and delete that element at critical value for $D$.

### 3.1.1. Uniform Corrosion Modeling

Uniform corrosion, as described earlier, occurs due to the uniform distribution of micro-galvanic cells in the material. As for external galvanic corrosion, both anodes and cathodes need to be in contact with a conductive medium such as water solutions. Thus, in finite element modeling, this type of corrosion is exposure mediated which means that elements on the exposed surface are only the elements subjected to corrosion. Uniformity of corrosion is applied by giving all the elements on the exposed surface the same probability of corrosion initiation (i.e., 100% probability) and having the same kinetics of corrosion as included in the Gastaldi et al. [29] and Grogan et al. [30] model using Equation (2).

$$\dot{D}_U = \frac{\delta_U}{L_e} k_U \tag{2}$$

where $\dot{D}_U$ represents the rate of evolution of damage factor. $\delta_U$ is a characteristic dimension for the uniform corrosion process (e.g., the critical thickness of the corrosion film). $Le$ is the characteristic length associated with the finite element type. For a brick element, $Le$ is defined as in Equation (3). It is positioned to be inversely related to the damage evolution to account for the element volume since a bigger element will degrade slower. Introducing this ratio $\delta_U/L_e$ is important to remove the dependency of the corrosion evolution on the mesh size. Furthermore, this ratio should be greater than one. This means that the finite elements sizes are guaranteed to be less than the smallest physical dimensions being modeled to ensure the accuracy. $k_U$ is related to the kinetics of uniform corrosion which represents the corrosion rate of the material [29].

$$L_e = \sqrt[3]{V_e} \tag{3}$$

$D_{cr}$ is not shown in the equations but it is also a major parameter that determines the onset of element deletion and the overall time to failure. Once the outer surface elements fail, the near elements are chosen according to their relative position to the failed element. In order to formulate this, Gastaldi et al. [29] used the concept of the radius of influence $\rho$. If neighbor elements are at a distance within that radius, they will be marked to start the process of corrosion again. This radius is calculated as in Equation (4) to account for the effect of the element size which is not constant in most cases when the mesh refinement is required.

$$\rho = \rho_0 * \frac{L_e^F}{\Delta} \tag{4}$$

where $\rho_0$ is a constant, $L_e^F$ is the characteristic length of the failed element, $\Delta$ is the maximum element dimension in the FE grid. Grogan et al. used a different method to update the corrosion surface which depends on the connectivity between elements (i.e., connected elements to the failed elements will be updated as corrosion surface) [30]. It is noteworthy that the model made by Grogan et al. included modeling of this type of corrosion just to compare it with the pitting corrosion model described later in this section.

In order to calibrate the model parameters, immersion tests are usually conducted to get experimental values using different corrosion measures explained briefly in the following sections. Gastaldi et al. [29] used in vitro immersion tests for circular disks of different Mg–Al–Zn alloys and the calibrated $k_U$ parameter was found in the range $10^{-2}$–$10^{-1}$ (hr$^{-1}$). $\delta_U$ was equal to 100 μm to model the reported corrosion product's thickness in reference [31]. This implies that the largest finite element in the model should have $L_e \leq 40$ μm to satisfy a maximum $L_e/\delta_U$ ratio of 0.4. The circular disks were modeled using a 2D finite element model with 4-node axisymmetrical elements with $L_e$ following Equation (5) [29].

$$L_e = \sqrt[3]{A_e 2\pi r} \tag{5}$$

where $A_e$ is the element area, $r$ is the distance between the point of symmetry to the centroid of an element $e$. Since these elements are two dimensional, it is not clear why they used this formula which is apparently the cubic root of the volume of a three-dimensional element. Grogan et al. [30] calibrated their model using foils of AZ31 with a much lower thickness of 0.21 mm in order to match the size of absorbable metallic stents which leads to different parameters summarized in Table 1. Material characteristic length $\delta_U$ was chosen to match the grain size of the AZ31 alloy of 170 μm [30]. Wu et al. [32] used the same model developed by Gastaldi et al. in [29] to study the effect of optimizing the stent material quantity for a maximum scaffolding time. They studied the corrosion time for three different stents' geometries by varying the size and number of peaks to valleys of the stents made of AZ31 alloy. Since the model includes the effect of stress corrosion cracking that will be discussed later, the optimized geometry showed the least stress concentration which led to the increase in scaffolding time even with less material amount than of maximum peak to valley number stents. They also showed that optimized stents can increase the scaffolding time with a minimum optimal material amount, which gives good evidence on the importance of having numerical simulations.

**Table 1.** Summary of uniform corrosion modeling parameters for the given materials.

| Research Group | Material | $\delta_U$(μm) | $L_{e,max}$(μm) | $k_U$(h$^{-1}$) |
|---|---|---|---|---|
| Gastaldi et al. [29] | AZ31, AZ61, AZ80, ZK60 and ZM21 | 100 | 40 | 0.01–0.1 |
| Grogan et al. [30] | AZ31 | 170 | 70 | 0.026 |
| Wu et al. [32] | AZ31 | 100 | 40 | 0.005 |

### 3.1.2. Modeling of Pitting Corrosion

Since pitting corrosion is almost inevitable in the case of magnesium alloys, it was investigated to extend the earlier uniform corrosion models. As discussed by references [28,30], the difference

between pitting modeling and the later model by Gastaldi is introducing a stochastic parameter $\lambda_p$ in the damage rate formula Equation (6). It is an idea first introduced by Wenman et al. [33] in studying the pitting corrosion that results in stress corrosion cracks. This parameter selectively chooses the first elements to initiate the damage based on a Weibull probability distribution function shown in Equations (7) and (8). Further, they introduced an acceleration parameter that is multiplied to the inherited $\lambda_p$ as in Equation (9), to account for the accelerated pitting corrosion and to ensure the development of randomly distributed pits around the initial conditions of pits nuclei.

$$\frac{\partial d_p}{\partial t} = \frac{\delta_u}{L_e} K_u \lambda_p \tag{6}$$

$$f(x : \psi, \gamma) = \begin{cases} \frac{\gamma}{\psi}\left(\frac{x}{\psi}\right)^{\gamma-1} e^{-(x/\psi)^\gamma}, & x < 0 \\ 0, & x \geq 0 \end{cases} \tag{7}$$

$$P\big[a \leq \lambda_p \leq b\big] = \int_a^b f(x)dx \tag{8}$$

$$\lambda_p' = \beta\lambda_p \tag{9}$$

where $\gamma$ and $\psi$ are the probability function constants, $\lambda_p'$ is the inherited probability value, $\beta$ is the acceleration constant, and *a, b* are the range values of the probability function.

Grogan et al. [30] compared the uniform corrosion model to the pitting model using experimental mass loss data and mechanical properties on foils of AZ31 alloy. Pitting model results showed better matching with experimental data in terms of mass loss, the change in tensile strength over time, and even the time to failure. Although they did not include the effect of SCC in the model, the pitting model showed better matching with experimental time to failure for samples with varied applied tensile stress, see Figure 5.

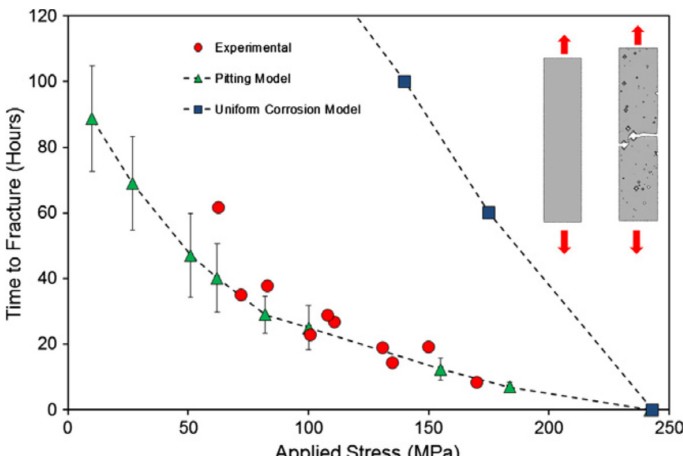

**Figure 5.** Shows the effect of applied stress on time to fracture for both pitting and uniform model, reprinted with permission from [30].

Furthermore, the pitting model showed localized attack and damage as compared to the uniform model as expected, see Figure 6. The probability density function parameters in Equation (7) $\gamma$ and $\psi$ are represented by their ration in the model by Grogan. Amerinatanzi et al. [28] used the pitting model by Grogan for a Mg–1.2Zn–0.5Ca (wt.%) alloy developed by the same group. Unlike Grogan et al., the response surface method was used to minimize the calibration runs and reach the optimum values for the model parameters. The four parameters had three levels values and 27 effective combinations were selected using Minitab v16 software. Mass loss data from immersion tests of the alloy coupons were used for comparison. The 27 runs were conducted on Abaqus/Explicit until the maximum experimental

mass loss was achieved. The set of parameters that resulted in the least error for this alloy are shown in Table 2.

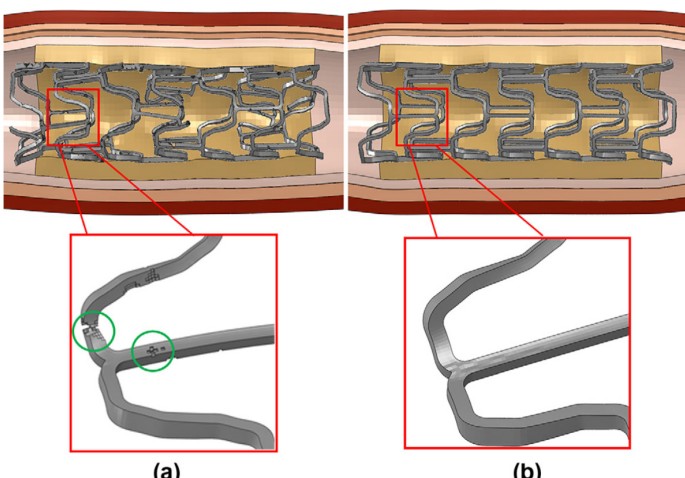

**Figure 6.** Comparison between pitting model in (**a**) and the uniform corrosion model in (**b**) by Grogan et al., reprinted with permission from [30].

**Table 2.** Summary of pitting corrosion modeling parameters for the given materials.

| Research Group | Material | $\delta_U$(μm) | $L_{e,max}$(μm) | $k_U$(h$^{-1}$) | $\gamma$ | $\psi$ | $\beta$ |
|---|---|---|---|---|---|---|---|
| Grogan et al. [30] | AZ31 | 170 | 70 | 0.00042 | 0.2 | – | 0.8 |
| Amerinatanzi et al. [28] | AZ31, Mg–Zn–Ca | – | – | 0.1005 | 2.748 | 2.60477 | 5.1 |

### 3.1.3. Modeling of Stress Corrosion Cracking

To model this phenomenon, several studies [29,32,34] used the formula for damage evolution as introduced by da Costa-Mattos et al. [35] for modeling SCC on stainless steel, see Equation (10).

$$\frac{\partial d_{sc}}{\partial t} = \begin{cases} \frac{L_e}{\delta_{sc}}\left(\frac{S*\overline{\sigma}_{eq}}{1-d_{sc}}\right)^R, & \overline{\sigma}_{eq} \geq \sigma_{th} \geq 0 \\ 0, & \overline{\sigma}_{eq} < \sigma_{th} \end{cases} \tag{10}$$

where $d_{SC}$ is the damage factor due to stress corrosion, $\overline{\sigma}_{eq}$ is the stress measure for controlling stress like Von Mises stress or any stress component. $S$ and $R$ are constants related to the kinetics of the stress corrosion process and can be a function of the corrosive environment. Unlike uniform and pitting corrosion, this type of corrosion is stress-mediated and the elements might not be subjected directly to the corrosive environment. Debusschere et al. [34] combined the model by Grogan et al. [30] and Gastaldi et al. [32] to study modifying the finite element model time integration method. They changed from an explicit to implicit integration method in order to minimize the time step increment, which optimizes the computational effort. They showed a good matching between their model and the previous models while using 1% of all degradation time as the time step.

In order to account for those different corrosion mechanisms, superposition principal is assumed applicable and the general damage factor is the summation of sub-types of damage factors. The most comprehensive approach was found to be by applying the effect of pitting corrosion with stress corrosion. In this case, the general $D$ is in Equation (11) by Debusschere et al. [34].

$$D = D_P + D_{Sc} \tag{11}$$

## 3.2. Physical Modeling

Although the phenomenological approach showed a good match with the experimental data, it fails to capture other aspects of the corrosion process. For instance, it cannot capture the interaction between dissolving ions of magnesium, corrosion products formation and dissolution, and the coating effects which were found necessary for corrosion tailoring and biocompatibility enhancement. Since the continuum damage approach is mainly considering the damage that occurs to the material by removing finite elements, it cannot model the generation of other elements to account for the formation of corrosion products. Such limitation of phenomenological modeling methods can be addressed using physical models. Furthermore, complex corrosion surface movement is applicable in physical models by using complementary tools such as the level set method [36].

### 3.2.1. Activation Controlled Modeling

Physical modeling of magnesium corrosion can be achieved in two ways [37]. The first is activation controlled which is a function in the potential difference between the anodic material and the solution resulting in a faster magnesium ions migration than the transformation of the magnesium into $Mg(OH)_2$ through the total electrochemical reaction (12). In this method, finite element or finite difference methods are used to solve a Laplace equation $\nabla^2 E = 0$ with appropriate boundary conditions for the scalar field of the potential difference $E$.

$$Mg + 2H_2O \rightarrow Mg(OH)_2 + H_2 \tag{12}$$

This potential difference drives the corrosion current of magnesium ions resulting in the movement of the corrosion boundary as adopted by Deshpande et al. [38]. The micro-galvanic corrosion was modeled internally between different magnesium alloy phases to study the effect of the percentage of the secondary phases $\beta$ and their distribution on the corrosion process. In the case of a continuous $\beta$ phase as in Figure 7, the average anodic current was found to increase with time as $\alpha$ dissolves, and the cathodic fraction increases up to a point when all the $\alpha$ is dissolved and $\beta$ is completely exposed to the solution. The corrosion rate then decreases as this phase has a lower activity. In a discontinuous $\beta$ phase, the average anodic current was found to be less than the continuous case because of the small cathode to anode surface ratio. However, the average mass loss over time is greater in this case because of the anticipated spatter that will happen when necking starts under $\beta$ particles. This phenomenon was reported while investigating the corrosion products of ingot alloy: $\beta$ particles were found in the product analysis [39]. Wilder et al. [40] used a finite difference with the level set method and adaptive mesh to model the external galvanic corrosion between mild steel and AE44 magnesium alloy. In these models, the corrosion surface speed is modeled as a function in the anode current, see Equation (13). The adaptive mesh introduced the ability of successive and localized refinements and coarsening to the mesh. In addition to increasing the accuracy, it is also important to solve the problem related to singularities at the corners where different boundary conditions coincide.

$$r_c = \frac{M}{zF\rho} i_{anode} \tag{13}$$

where $r_c$ is the corrosion rate or the speed of the corrosion surface, $M$ is the atomic mass of the anodic metal or alloy, $\rho$ its density, $z$ the electron number, $F$ Faraday's constant, and $i_{anode}$ the current density of the galvanic couple at a specific point on the anode.

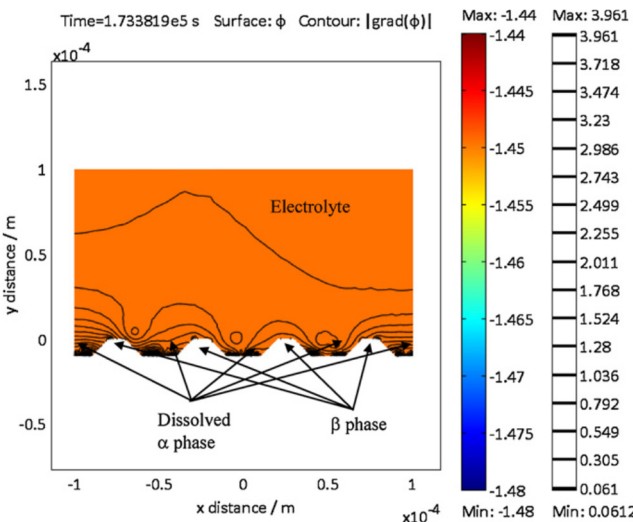

**Figure 7.** Model results for a continuous $\beta$ phase network at time $t = 1.73 \times 10^5$ s where the surface plot indicates the potential field and contour represents absolute potential gradient, reprinted with permission from [38].

Montoya et al. [41] used this method with an in vivo experiment to study the effect of the electrolyte amount around an implanted magnesium rod inside a Wistar rat femur bone. For cast magnesium, this amount seems to affect the corrosion potential unlike AZ31 alloy. They found in a 2D finite element model that the radial thickness of the electrolyte below 0.42 mm would decrease the corrosion potential substantially which translates into less corrosion for the case of cast magnesium. In vivo CT images show no formation of gaseous products at the middle of the implant where electrolyte amount is near zero.

In such models, parameters to be selected are the electrolyte/solution/surrounding medium conductivity, the anodic current function in the applied potential $i_{anode}(E)$ of the anode metal, and the cathodic current function $i_{cathode}(E)$ if it is not considered constant of the cathode metal or phases. These functions can be approximated as piecewise linear functions from the experimental polarization nonlinear curves [40,41].

### 3.2.2. Transport-Controlled Modeling

The potential applied, which represents the activation effect, is the dominant factor at the very beginning of corrosion. Meanwhile and in the long run, the corrosion was found to be independent of the activation effect [42]. Increasing the voltage no longer increases the current since a passive corrosion product precipitates on the surface restricting the ions from migration to the solution. Furthermore, in an in vivo scenario, there are layers of the tissues surrounding the implant, reducing the voltage difference effect. Hence, the corrosion process can be considered transport-controlled instead, since species will have to flow through the corrosion surface and any deposited layers on the surface (e.g., tissue cells, coating, etc.). Transport of species $i$ can be represented by Nernst–Planck Equation (14) [38] as:

$$N_i = -D_i \nabla c_i - z_i F u_i c_i \nabla \varnothing + c_i U \tag{14}$$

where $N_i$ is the flux; $D_i$ is the diffusion coefficient; $c_i$ is the concentration; $z_i$ is the charge and $u_i$ is the mobility of species $i$, respectively; $F$ is the Faraday's constant; $\varnothing$ is the potential and $U$ is the solvent velocity. The first term represents the effect of the diffusion on the flux, second term is the galvanic corrosion effect, and the third is the convection effect. Figure 8 shows a schematic of the migration of ions under the diffusion effect.

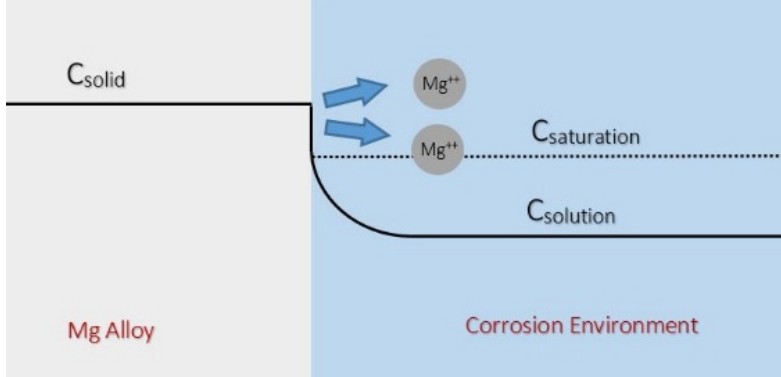

**Figure 8.** Schematic of the assumed corrosion process. Mg++ ions dissolve into the solution which has solubility limit of $C_{saturation}$. As the Mg++ ions dissolve, the boundary moves inwards with velocity v in normal direction to the boundary.

From the conversation of flux of species *i* the following equation applies [38]:

$$\frac{\partial c_i}{\partial t} = -\nabla \cdot Ni \tag{15}$$

Under the assumption that the corrosion is mainly transport-controlled and substituting from Equation (14) into (15), the following relation reveals for magnesium ions concentration:

$$\frac{\partial c_{Mg}}{\partial t} = \nabla\left(D_{Mg} \cdot \nabla c_{Mg}\right) \tag{16}$$

Equation (16) is referred to in literature as Fick's law and was used by Grogan et al. [37], Scheiner et al. [42], Gartzke et al. [43], and Bajger et al. [36] while the latter added the effect of the corrosion product formation and degeneration to the equation. This equation is based on three main parameters that control the transport process [42]. The first is the concentration of the magnesium in the solid material which can be considered as the density in the case of a pure magnesium. The second is the saturation concentration of the magnesium into simulated body fluid (SBF) under particular pH and body temperature. The last is the diffusivity of the magnesium ions into the SBF solution. The last two parameters need further experimental work since they are very controversial in literature as per reference [37].

Knowing the governing equation for corrosion, now the movement of the corrosion surface was investigated in the previous studies. For Grogan et al. [37] and Scheiner et al. [42], the moving boundary method was used. Figure 9 shows the simulation results obtained in reference [37] using the moving boundary method. A method was originally developed to model the ice/water transformation in polar seas. For the other two groups, the level set method was used for its ability to follow the complex topology of the corroding surface. It is noteworthy that the Grogan et al. [37] group is also working on this method as they mentioned at the end of their paper while taking the effect of the corrosion product formation into consideration.

The work done by Bajger et al. [36] can be considered an extension to the work done by Grogan et al. [37]. In their model, they added the effect of the chemical reaction that happens at the corrosion surface resulting in deposition and dissolution of the corrosion products to better simulate the real scenario. While in Grogan model, they assumed the presence of the corrosion products in the very first time interval and their model was for long-term corrosion. A set of coupled differential Equations (17)–(20) was assumed to control the corrosion process which added more parameters to calibrate. Equation (17) relates the $c_{Mg}$ to the corrosion product's thickness denoted by F and the

chlorine ions concentration. Up to the first term in the equation, it is typically the same as the model by Grogan et al.

$$\frac{\partial c_{Mg}}{\partial t} = \nabla\left(D_{Mg}^e \cdot \nabla c_{Mg}\right) - k_1 c_{Mg}\left(1 - \frac{F}{F_{max}}\right) + k_2 F c_{Cl}^2 \tag{17}$$

Second term is negative and is valid as long as $F \leq F_{max}$; the negative sign means it has reducing effect on the rate of change in $c_{Mg}$ with a relation constant $k_1$. Additionally, the rate of change is increased by the diffusion term and the degeneration of the corrosion product represented as function in the $c_{Cl}$ presence and has the rate constant $k_2$.

Equation (18) is about the rate of formation of the corrosion film and it is function in $c_{Mg}$ and $c_{Cl}$.

$$\frac{\partial F}{\partial t} = k_1 c_{Mg}\left(1 - \frac{F}{F_{max}}\right) - k_2 F c_{Cl}^2 \tag{18}$$

Finally, the rate of change in $c_{Cl}$ as function in its diffusivity which needs further calibration.

$$\frac{\partial c_{Cl}}{\partial t} = \nabla\left(D_{Cl}^e \cdot \nabla c_{Cl}\right) \tag{19}$$

The relation between corrosion products formation and slowing down the diffusion is interpreted in relating the diffusivity factor to the corrosion film thickness which is called the effective diffusivity. Assuming the corrosion product is a porous medium, Equation (20) was introduced by Bajger et al. [36] to calculate the new effective diffusivity at each time step.

$$D_c^e = D_c\left(\left(1 - \frac{F}{F_{max}}\right) + \frac{F}{F_{max}}\frac{\epsilon}{\tau}\right) \tag{20}$$

where $D_c^e$ is the effective diffusivity of species $c$, $\epsilon$ is the porosity and tortuosity $\tau$ of the assumed layer of Mg(OH)$_2$, and $D_c$ is the free diffusivity at zero corrosion film thickness. Tortuosity of a porous medium is defined as the ratio of the average actual flow path length to the straight distance between the medium start and end [44].

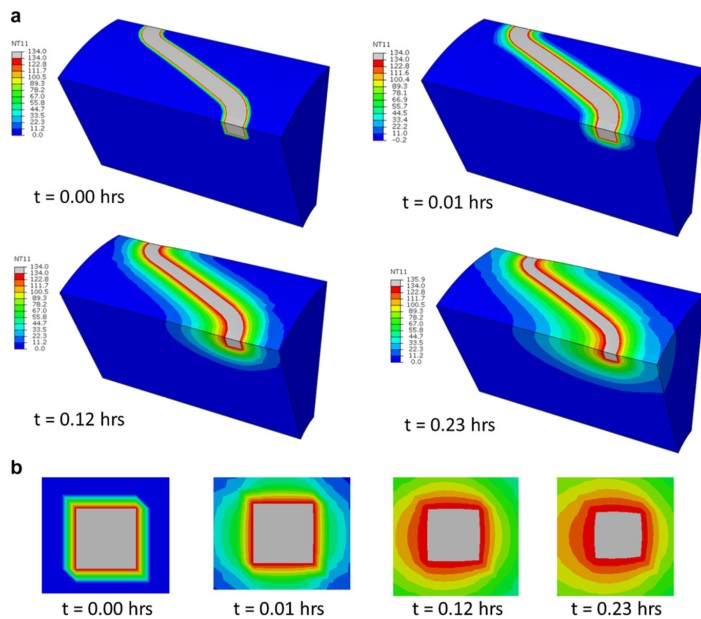

**Figure 9.** Contour plots of the magnesium concentration distribution in the model by Grogan et al., reprinted with permission from [37]. (**a**) The upper half of the model has been removed for illustrative purposes. Grey regions correspond to non-corroded metal. (**b**) Illustration of the changing dimensions of a cross-section of the hinge as it corrodes.

Modeling of stress corrosion cracking that is induced by hydrogen embrittlement of AZ91 alloy is investigated by Dietzel et al. [45]. Since the fracture is assumed to be a result of hydrogen accumulation in corrosion pits, a mesoscale fiber bundle model is introduced in which failure is assumed as the successive failure of parallel fibers to the applied load. The hydrogen embrittlement is simulated as the reduction in the critical strain at failure as a function in the hydrogen concentration in the fiber. The critical strain at failure in the air is measured and Equation (21) holds this relation.

$$\varepsilon_f = \varepsilon_{f0} \exp(-x_H u_H) \tag{21}$$

where $\varepsilon_f$, $\varepsilon_{f0}$ are the critical strains at the failure of the material when deformed in an aqueous environment and in air, respectively. The factor $x_H$ is a numerical constant, $u_H$ is the normalized hydrogen concentration (i.e., $u_H = C_H/C_0$), $C_0$ is the normalizing hydrogen concentration. Now Equation (21) is a function in hydrogen concentration. Finite difference method was used to solve for this concentration in a two-dimensional model. Boundary conditions for the concentration were put to capture the stochastic nature of the pits' initiation, thus random elements on the outer surface are randomly selected to have unity hydrogen concentration to start the diffusion problem. Hydrogen diffusivity was determined to be $D_{eff} = 2x10^{-13} \; m^2 S^{-1}$ and $x_H = 5$ to best match experimental results.

### 3.2.3. Modeling of Coating Effect

Magnesium is often subjected to alloying, coating, or mechanical processing to limit the degradation rate to safe levels inside the body [46,47]. Different coating techniques were reported in the literature to improve the corrosion resistance of magnesium alloys [48–50]. In order to model such an effect, coating can be represented by reducing the diffusivity at the implant boundary as compared to uncoated alloy using a similar equation to Equation (20). In addition, interaction or degradation of such coating can be considered in an analogous manner as discussed in Bajger et al. [36]. Figure 10 shows the results by the latter group in modeling the corrosion products distribution on the surface. As mentioned earlier, modeling of such effects distinguishes the physical modeling approach over phenomenological models.

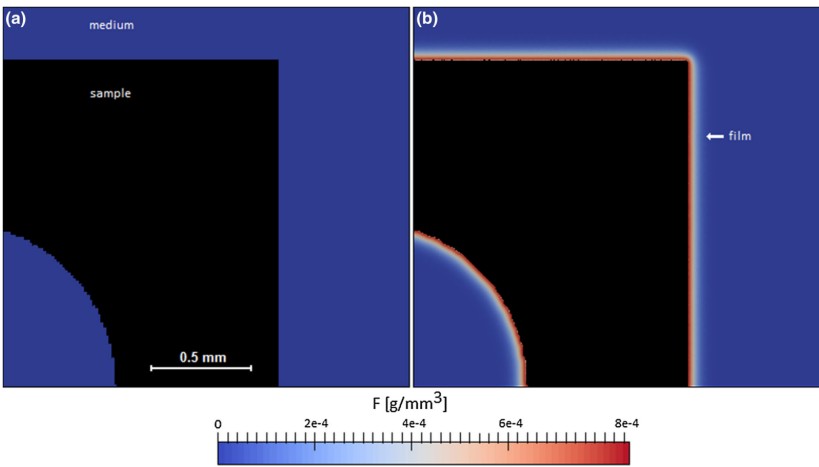

**Figure 10.** *F* represents the corrosion products concentration on the sample surface, (**a**) at time *t* = 0 and (**b**) at *t* = 7 days of immersion, reprinted with permission from [36].

### 3.3. Cellular Automata Corrosion Modeling Approach

Another method that combines the phenomenological and physical methods is based on the cellular automata (CA) approach. To the best of the authors' knowledge, this method has not been investigated for modeling biodegradable metals but will be reviewed in this section for its high potential for future research. Pidaparti et al. [51] used this method to model pitting corrosion on aircraft aluminum on a two-dimensional grid. For a summary of the complex physical systems that can be

modeled using cellular automata refer to reference [51]. In this method, the n-dimensional domain is discretized into cells while each cell carries a "state" variable. Each cell has a neighborhood of cells that interacts with it. There are several neighborhoods that vary in size to represent how localized the interaction is. The evolution of each cell state is a function in the selected neighborhood cells states which is represented by a "rule" in the modeling code. Thus, starting from initial conditions at $t_0$, subsequent states can be calculated [52]. Since pitting corrosion or corrosion, in general, is due to an interaction between metal phases, this method can be used by correctly selecting the right rules for transitioning from time $t$ to $t + 1$.

CA combines the properties of phenomenological models where the corrosion surface tracking is not required. Corroded elements can just be hidden at a threshold value of state which is similar to the continuum damage model with a scalar field of a damage factor. It shares the ability to capture the physical interaction between the metal and the solution with the physical modeling methods. Furthermore, CA does not require heavy finite element or finite difference calculations, hence, higher resolution and smaller time steps can be achieved.

Pidaparti et al. [51] developed a 2D simulation of the random pitting corrosion in two steps. The first step or sub-model is the pitting initiation model in which each cell $u$ is assigned an initiation state $I(u,t)$. At each time step, the initiation state is increased with a random number to represent the probabilistic nature of pitting. Once $I(u,t)$ exceeds a threshold value $H$, the corrosion sub-model begins with assigning a low corrosion state $S(x,t) = 3$:5 on the range from 0 for uncorroded to 255 for completely corroded. They defined the rule for state development over time as function in the relative location of neighbor cells to the central cell, the electrochemical properties, and the environment properties from pH and temperature which represents how this method is similar to physical modeling.

Di Caprio et al. [53] introduced a general two-dimensional CA-based model for a combined uniform and pitting corrosion for metals. Furthermore, the model can capture the formation of a passive layer that affects the corrosion rate via diffusion of corroded metal grains and precipitation on the surface. The model is a function in four simple parameters: (i) a parameter that represents the ratio of the diffused corrosion products to the standstill potion on the surface, (ii) a single parameter that represents all the kinetics of the metal corrosion and can be metal-specific, (iii) a parameter "$\lambda$" that differentiates between the uniform corrosion region with a lower corrosion rate and the bottom region that represents pitting localized corrosion with a higher corrosion rate, and (iv) a parameter "$\varepsilon$" that modulates the corrosion rate in the two regions. Holding the first two parameters constant and tweaking the last two parameters reproduced a complex morphology that qualitatively matches experimental images, see Figure 11. In later work by the same group, Stafiej et al. [54] used a two-dimensional CA model to illustrate the pit formation due to the depassivation in the surface layer. Development rules were put to account for the repassivation and dissolution of the pit surface. Phenomenon like peninsula formation and islands detachment could be observed as in reference [54].

Lishchuk et al. [55] developed a CA model for the intergranular corrosion process based on four parameters and two sets of rules that govern the corrosion rate or the movement of the corrosion surface. The simulated metal is modeled as a brick wall, where bricks represent the grains, and the mortar represents the grain boundary phases. The metal is covered by a layer of passive film that is also assumed to corrode. The first three parameters of the model are probabilities of corrosion of the different structure elements namely grains, grain boundaries, and surface layer. The intergranular corrosion was modeled by giving the grain boundary cells the top corrosion probability. The fourth parameter is the time step. Since the time step is not implicit in the model development equation (e.g., CD models), it has to be calibrated as a dependent parameter in CA models to fit the experimental results. The dissolution of corrosion products was modeled by random walks in the program in which the corrosion products were allowed to migrate in random paths until they disappear in the solution.

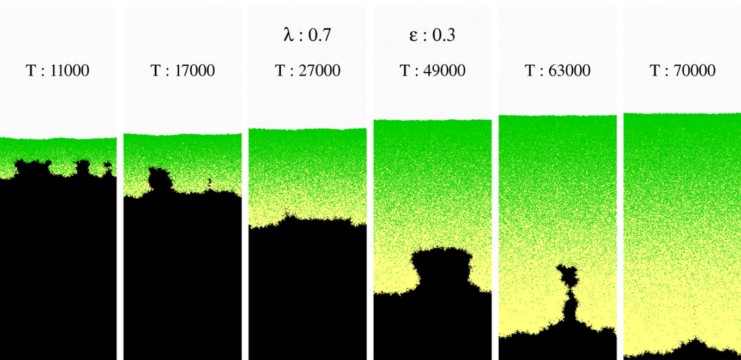

**Figure 11.** Time snapshots of the 2D model by Di Caprio et al., $\lambda = 0.7$ and $\varepsilon = 0.3$, reprinted with permission from [53].

Di Caprio et al. [56] expanded their model to a more general three-dimensional CA model to study the intergranular and transgranular corrosion. The same concept of probabilities as in reference [55] is used to study the effect of the geometry and distribution of the grains on the corrosion rate. Using the concept of probabilities could help reduce the time steps required by increasing the probability of corrosion [56]. Grains were modeled as cubes, bricks, and random shapes generated by Voronoï tessellation to study the effect of grains geometry. More random grain shapes can force the intergranular corrosion path to elongate and change direction rather than just follow the normal direction to the corrosion surface as with cube and brick-shaped grains. Generally, the corrosion rate was found faster using the 3D model than the previously developed 2D models by the same group [56].

## 4. Corrosion Surface Tracking: Level Set Method

The level set method is widely used in several applications and it was first introduced by Osher and Sethian in the late 1980s to model the propagation of the interfaces between two moving media [57]. For a good introduction to this topic, one can refer to these introductory books [58,59]. Since the problem of corrosion involves the movement of the corrosion front while following the shrinking metal, some research groups used this method to track the corrosion front movement over time to predict the resulting topology in any space dimension [26,36,38,40,60]. In Figure 12, the corrosion surface denoted by $\Gamma$ is given the zero contour level at each time step by the time dependent and distance function $\varphi(x, t)$ after solving the hyperbolic and nonlinear PDE Equation (22).

$$\frac{\partial \varphi}{\partial t} + v \cdot \left| \nabla \varphi \right| = 0 \tag{22}$$

where $x$ is the location vector of a point and $v$ is the propagation velocity. The process of solving this equation involves two main steps: (i) discretization by a numerical method which can be finite difference method [26,40] with a good accuracy, and then (ii) re-initialization of the distance function at the new positions. As the boundary evolves over time, the distance function starts to deviate from the distance function property that $\left| \nabla \varphi \right| = 1$, which deteriorates the solution accuracy in further time steps [61]. Therefore, the classic solution of this method always involves a re-initialization step to maintain accuracy. The moving boundary divides the domain $\Omega$ into two subdomains ($\Omega_+, \Omega_-$); each domain contains the points of the shortest signed distance from the boundary where positive values can be assumed inside the closed boundary (i.e., the solid metal). As shown in Figure 12, each point on the boundary is assumed moving in the opposite direction of the normal vector at that point with a defined speed $v$. The movement speed is a function in the corrosion kinetics that were assumed in the model. In the case of activation controlled models, $v$ takes the same value as $r_c$ in Equation (13) [40]. In the transport-controlled case, $v$ is a function in the diffusivity and concentration gradient of magnesium ions [36,37]. In this method, complex emerging or separating boundaries can be handled easily which helps to model complex geometries [57]. Figure 13 shows the results obtained by Bajger et al. to model

the 2D movement of the corrosion surface for pure magnesium, which shows the capability to model the separation of surfaces. In the Bajger et al. [36] model, Equation (23) was used to solve for the signed distance function. The second term coefficient is a function in the diffusivity and different magnesium concentrations and concentration gradient.

$$\frac{\partial \varphi}{\partial t} - \frac{D_{Mg}^e \nabla_n c_{Mg}}{c_{Mg(sol)} - c_{Mg(sat)}} |\nabla \varphi| = 0 \tag{23}$$

where $D_{Mg}^e$ is the effective diffusivity of magnesium ions, (calculated from Equation (20)) and $\nabla_n c_{Mg}$ the magnesium ions concentration gradient, calculated at distance $h$ from the corrosion surface, where $h$ is the shortest element radius in the mesh [36]. $c_{Mg(sol)}$ is the magnesium ions concentration in the solid metal and equals the density of magnesium in the case of pure magnesium [37], $c_{Mg(sat)}$ the saturation limit of the solution and was approximated by the saturation limit of magnesium chloride in water at 25 °C by Grogan et al. [37] due to the unavailability of data on this limit in appropriate physiological parameters.

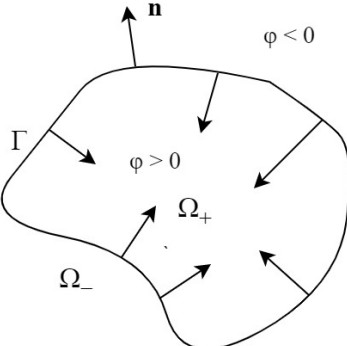

**Figure 12.** Corrosion front as represented by level set method.

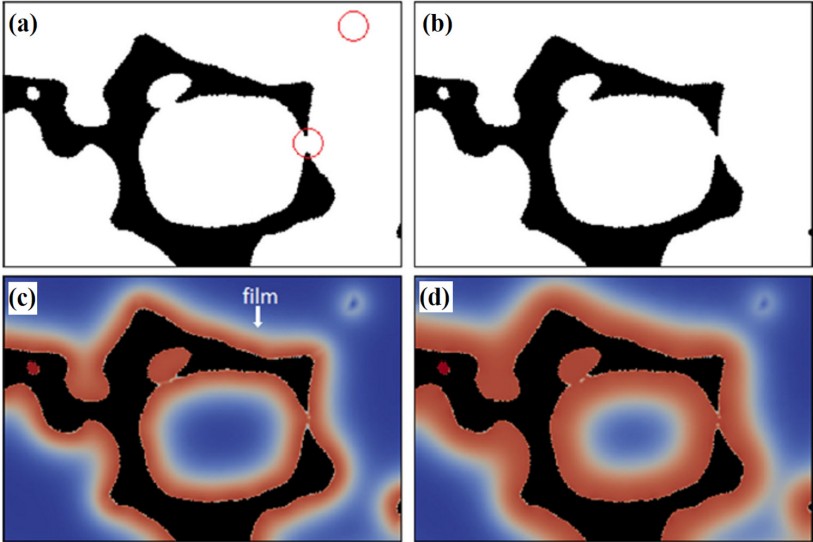

**Figure 13.** (**a**,**b**) shows the change in topology after 7 and 35 days of immersion, respectively. (**c**,**d**) shows the predictions of the topology by the LSM by Bajger et al. after 14 and 28 days respectively, reprinted with permission from [36].

## 5. Calibration Test Methods

All the modeling methods mentioned in the previous sections are dependent on some parameters that require experimental calibration in order to match the model predictions to the experimental

results. In addition, experimental data is essential to validate the results obtained from any developed model. This section provides a review of the experimental practices that have been used in the literature to calibrate/validate corrosion modeling attempts of biodegradable metals.

### 5.1. In Vitro Testing

In vitro refers to the testing methods that are performed outside of a living body in an environment simulating the physiological conditions. In the case of an in vitro corrosion (degradation) testing, simulations are typically performed using a simulated body fluid such as the Hanks' balanced salt solution (HBSS), which has a balanced mixture of inorganic salts that approximate the balance of those same salts found in the human body fluids and is typically used in most experimental procedures for testing in vitro degradation [15]. For a comprehensive review of the effects of different solutions the reader can refer to reference [62]. There are two primary focuses for in vitro testing: (i) biodegradation/corrosion rate, and (ii) toxicity/organism impact. If a magnesium implant degrades too quickly inside of the body, the reconstructed bone will not heal properly and the increased concentration of released irons due to corrosion can cause harm to the surrounding tissues and other organisms [63–68]. Due to its importance, the degradation rate serves as the main parameter to calibrate numerical models included earlier.

The in vitro testing can be split into polarized and unpolarized methods of testing. The in vivo testing can be performed in animals, or in humans when clinical trials have been cleared in the field of research. There has never been a strong correlation relating in vitro and in vivo testing results. However, the information collected from in vitro tests can provide important insights that can lead researchers when preparing for in vivo studies.

### 5.1.1. Unpolarized tests

### Mass Loss

Measuring the mass loss (ML) is a common approach to assessing the corrosion behavior of biodegradable metals. In this approach, small samples of the tested material with known weight (e.g., pure magnesium or its alloys) are placed in the test fluid (e.g., HBSS), and degradation is allowed to occur for a period of time. The samples are generally prepared according to the standard sample preparation for immersion tests—ASTM G31-72 [69]. Test temperature, as one of the main physiological parameters, is kept at $37 \pm 0.5$ °C by placing the experiment setup inside a thermal incubator [15]. Due to the nature of magnesium corrosion in aqueous solutions, the pH level in the solution increases as magnesium reacts with water, therefore, titration with diluted HCl [19] or 5.96 g/l HEPES solution [15] is recommended to keep the pH level at $7.4 \pm 0.05$ °C. The sample is then removed from the fluid, cleaned according to the ASTM G1-03 standard practice [70] and measured on a scale. This method is low cost and yields an accurate measure of the material being lost assuming that no extra material is lost when removing the corrosion products [15]. The process of ML depends on the interaction between magnesium and the hydroxide ions in the fluid, per Equation (12). It is a very common method of establishing corrosion when performing in vitro tests [15,63–65,67,71–77]. ML is then measured in units of mass per exposed surface area, see Equation (24).

$$ ML = \frac{m_i - m_f}{A} \tag{24} $$

where $m_i$ is the initial sample mass, $m_f$ is the final mass after corrosion, and $A$ is the exposed area of the sample to the solution.

One of the alternatives to HBSS was used by Xu et al. [72], where a physiological saline solution (0.9% NaCl at a pH of 7) was used. Magnesium samples were immersed in the HBSS solution for 30 days, and the degradation was observed through the crack propagation in coated and uncoated samples. The samples were coated using steam oxidation and micro-arc oxidation. The concentration of

magnesium ions in the solution was determined using inductively coupled plasma mass spectrometry (ICPMS) as well as measuring the pH balance. A similar method was used by Yfantis et al. [76], with magnesium being submerged in a neutral pH saline solution; the degradation rate was determined by the mass loss at the end of the testing (22 days). The degradation rate was further compared in this study to the rate established using electrochemical corrosion testing, and it was found that the approximate corrosion rate calculated from the electrochemical corrosion test was 3–5 times higher than that calculated using ML by immersion.

Hydrogen Evolution Measurement

Hydrogen evolution is another method to measure the degradation rate that was used by several research groups [63,64,66,74,76,77]. The degradation of magnesium inside aqueous solutions was found to result in hydrogen gas evolution according to Equation (12). By this equation, for every mole of magnesium interacting with water, one mole of hydrogen gas is generated [15]. The setup is very similar to mass loss, with the sample of magnesium being submerged in the solution and a gathering mechanism affixed above the sample to collect the released hydrogen gas ($H_2^{Evo}$). This setup often uses a funnel and a burette, both filled with the corrosion fluid, settled over the sample to be tested, as demonstrated in Figure 14.

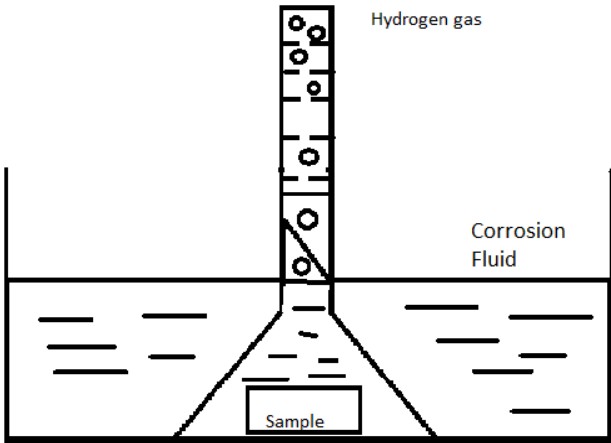

**Figure 14.** Setup for hydrogen gas collection with the funnel and burette for hydrogen gas capture.

Hydrogen gas forms from the interaction between the magnesium and the surrounding solution then rises to be collected in the capture burette. The fluid is displaced and pressed out of the tube back into the fluid. Hydrogen molecules are then calculated by volume, and the previously established equation gives an indication of the amount of magnesium that has reacted with the aqueous solution, as in Equation (12). There are several limitations associated with this method of evaluating the degradation rate as discussed by Kirkland et al. [15]. For example, the data calculated from the hydrogen evolution was always different than the ML data and a ratio of 1:1 between them could not be achieved. This can be attributed to several reasons such as the difficulty of capturing all the evolved hydrogen gas due to the reaction. In addition, the atmospheric pressure may need to be adjusted based on the altitude of the experiment. Another limitation of this method is the lack of accessibility to titrate the solution by buffers to adjust the increased pH value [15]. The "area of effect" of the test solution is considered the area of the corrosive fluid inside of the funnel. This setup has a "closed" system, with the funnel touching the base of the container, restricting the flow of the solution away from the magnesium sample. As a result, the pH change within a period of time is significantly different in the local area beneath the funnel from the case where the sample is interacting with the whole system.

In Figure 15 the magnesium sample is placed into a corrosive environment and allowed to degrade for the desired period, with one of the systems being an "open" system, where the corrosive fluid can

move freely (250 mL of corrosive material), and the second being a hydrogen evolution system, with a 50 mL funnel placed over the magnesium sample. The pH change in the smaller system occurs not only more rapidly, but to a higher level inside the system. However, the hydrogen gas evolution was found effective to give a relative comparison between two different alloys in terms of corrosion rate. Equation (25) is used to calculate the change in the sample weight based on the hydrogen gas being generated [15].

$$\Delta W = \frac{1.085\ V_H}{P_{ATM}} \tag{25}$$

where $\Delta W$ is the change in mass (mg), $V_H$ is the volume of the evolved hydrogen (mL), and $P_{ATM}$ is the atmospheric pressure (atm) [15,63,64,66,74,77].

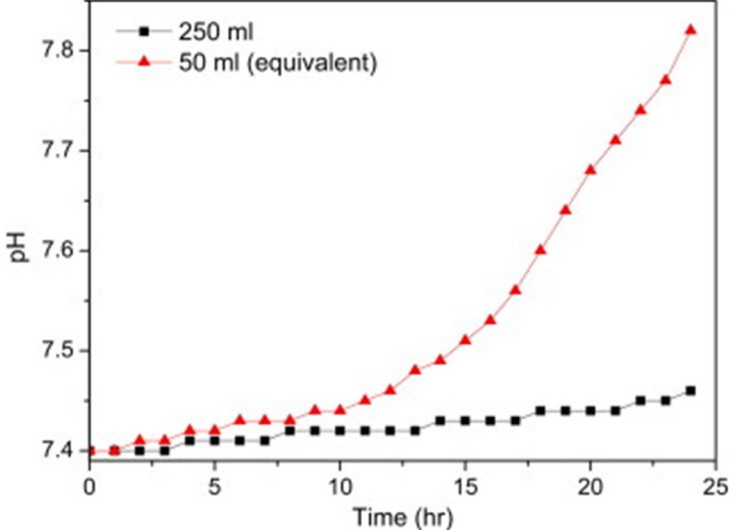

**Figure 15.** Demonstration of how the pH changes in the system for two different volumes, reprinted with permission from [15].

pH Monitoring

Corrosion of magnesium in aqueous environments increases the solution pH number due to the liberation of hydroxide ions in the solution [22]. Measuring the pH increase can provide some insights when comparing the corrosion rates of two different metals or testing the effect of modifications on the corrosion resistance [78–80]. However, monitoring the pH level does not seem to be a reliable approach to give a quantitative rate of corrosion for the tested sample. This can be attributed to the fact that a pH of 7.4 is an important physiological condition to maintain in order to simulate the in vivo environment when assessing the corrosion behavior of magnesium [22].

5.1.2. Polarized (Electrochemical) Method

Potentiodynamic Polarization

Often referred to as PDP, this method is the most commonly used electrochemical method of testing the degradation of samples. To start, the sample to be tested is brought to a steady or near steady state at the open circuit potential (OCP). After OCP is established, voltage is applied between the magnesium and an inert metal electrode. The initial voltage is set to a more negative, or cathodic, value to the OCP, then the voltage and scan are shifted to more positive, or anodic, values compared to the OCP [15]. The testing takes approximately five minutes and provides information on corrosion potential ($E_{corr}$), the reaction of magnesium with anodic and cathodic voltages, and kinetic information

from the corrosion current density ($i_{corr}$) [15,64,72,75,76]. The corrosion rate is then established based on Faraday's law using Equations (26) and (27) [81].

$$CR = 0.00327 \frac{EW \cdot i_{corr}}{\rho} \tag{26}$$

$$EW = \sum \frac{f_i \cdot a_i}{n_i} \tag{27}$$

where $CR$ is the corrosion rate (mm/year), $i_{corr}$ is the calculated current density ($\mu A/cm^2$), $\rho$ is the density ($g/cm^3$), $EW$ is the equivalent weight in the case of alloys, $f_i$ is the mass fraction of element $i$, $a_i$ is its atomic weight, and $n_i$ is the number of valence electrons of each element $i$.

Samples for PDP testing can be used more than once, as long as the sample has been cleaned and previous corrosion products have been removed from the sample before a new test is performed. The information gathered from this method of testing can provide an outline of the anodic/cathodic degradation differences in materials that have similar corrosion current density. In this review [15], Figure 16 described how samples with similar $i_{corr}$ can have different $E_{corr}$.

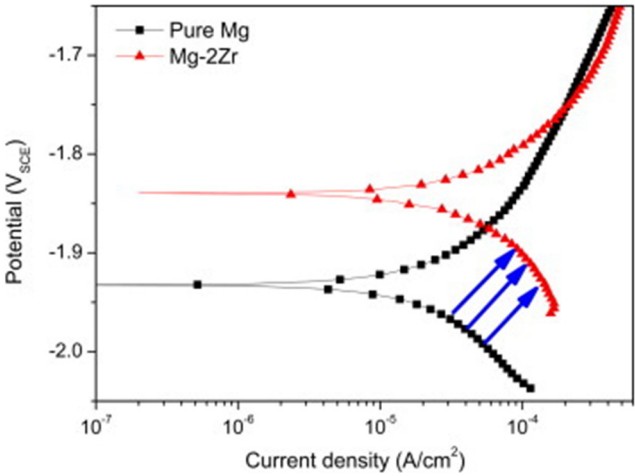

**Figure 16.** Polarization curves for pure Mg and Mg–2Zr. Arrows are used to indicate the cathodic shift (Hanks' balanced salt solution (HBSS), pH 7.4, T_phy), reprinted with permission from [15].

Magnesium alloys do not corrode uniformly, and the conversion of $i_{corr}$ to a corrosion rate can only be performed if corrosion is assumed to be "general corrosion" [15]. With this in mind, PDP cannot be used to establish an absolute corrosion rate for magnesium samples but can display the severity of corrosion at the desired point in time in relation to the current density. Most magnesium alloys do not corrode uniformly in practice, and there are no tests currently that provide an absolute prediction on the corrosion rate [15,75,76,82].

Electrochemical Impedance Spectroscopy (EIS)

This method uses the frequency response of AC polarization [15,64] and the low magnitude polarizing voltages in a cycling pattern from the peak anodic to peak cathodic voltages along with varied frequencies. The result is a set of values with instantaneous data on the impedance of a surface when it is subjected to polarization. The impedance is directly proportional to the corrosion resistance and inversely proportional to the corrosion rate. It can also be used to determine the dissolution and degradation rate when it is occurring. This method of testing is non-destructive when used with magnesium in an SBF, and a single sample can be tested multiple times without reworking the sample. Each sample can also have real-time, on-line monitoring [15,64,82,83].

EIS is useful when observing the relationship between the coating on magnesium before corrosion has been introduced and when the coating layer and magnesium start to break down. The EIS data can be paired with Tafel slopes measured by PDP data to obtain an approximate $i_{corr}$ value, but this requires the use of the Stern–Geary equation and relies heavily on accurate determination of the Tafel slopes of the individual reactions [82]. Having the surface resistance $R_p$ and calculating $B$ from Tafel slopes derived from the PDP reaction data, $b_a$ (anodic reaction) and $b_c$ (cathodic reaction) with Equation (28), $i_{corr}$ can be calculated from Equation (29).

$$B = \frac{b_a * b_c}{2.3\,(b_a + b_c)} \tag{28}$$

$$R_p = \frac{B}{i_{corr}} \tag{29}$$

where $R_p$ is the polarization resistance, $i_{corr}$ is the corrosion current, and $B$ is the proportionality constant for a particular system.

EIS, as in Figure 17, only provides some of the information required for corrosion kinetics, and cannot be used to determine $E_{corr}$ of the reaction caused by different alloying elements or solutions [15]. In addition, EIS does not directly yield a corrosion rate and is susceptible to degradation itself while the scan is running. As this degradation occurs, low frequency scans become more and more difficult to run, as the active reactions continuously occur while the impedance/resistance is recorded as steady corrosion continues [15,82,83].

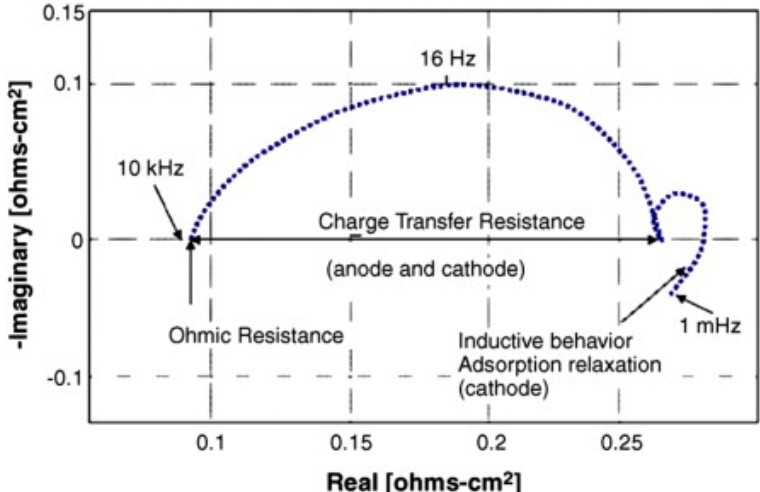

**Figure 17.** Electrochemical impedance spectroscopy (EIS) reaction, displaying the charge difference between the anode and cathode reactions with battery testing, reprinted with permission from [83].

## 5.2. In Vivo Testing

In vivo testing refers to testing processes that involve implanting the tested sample material inside a live body to observe how it reacts. In this process of testing, the model organisms being used are often rats, mice, or rabbits [65,67,71,75]. The treated magnesium sample is inserted into the test subject, in a subcutaneous sense [67,75], into muscle [67], or into bone [65,67]. In vivo testing uses a sample blank (a disc or cylinder shape) that is inserted into the animal and allowed to degrade [15,64–67,72,74,75,84]. Regular visual degradation is observed typically using an X-ray method or micro CT scan. As previously stated, a direct relationship between in vivo and in vitro testing has not been established, so performing in vivo testing provides real results in a real environment instead of a simulated environment.

The results from one such set of testing [74] used three groups of rats ($n = 5$, per group) with implants of (i) magnesium, (ii) magnesium with sodium montmorillonite (MMT) coating, and (iii) magnesium with an MMT/bovine serum albumin (BSA) composite coating in the individual groups. A

scan was taken 24 h after the implant, and again 120 days after the implant, displayed in Figure 18 below (a1,a2,b1,b2,c1,c2). After the scan, the rats were sacrificed, and the implants were removed for further analysis on degradation levels in the sample itself. Histological analysis was performed on various body parts to determine the impact of the magnesium rod after 120 days of implantation. The figure from the energy dispersive spectroscopy (EDS) scan Figure 18a5,b5,c5 demonstrates the impact of the inserted pins on the animal's physiology. In Figure 18b5,c5, the presence of calcium in the scan indicates the coating layer on the tested pins is diffusing out into the bloodstream of the test subject. Under the SEM images, the degradation of the surface of the pins can be seen, where magnesium displays significant cracking and degradation, Mg–MMT and Mg–MMT/BSA show comparatively fewer cracks to indicate the coating is not degrading as significantly as the magnesium itself.

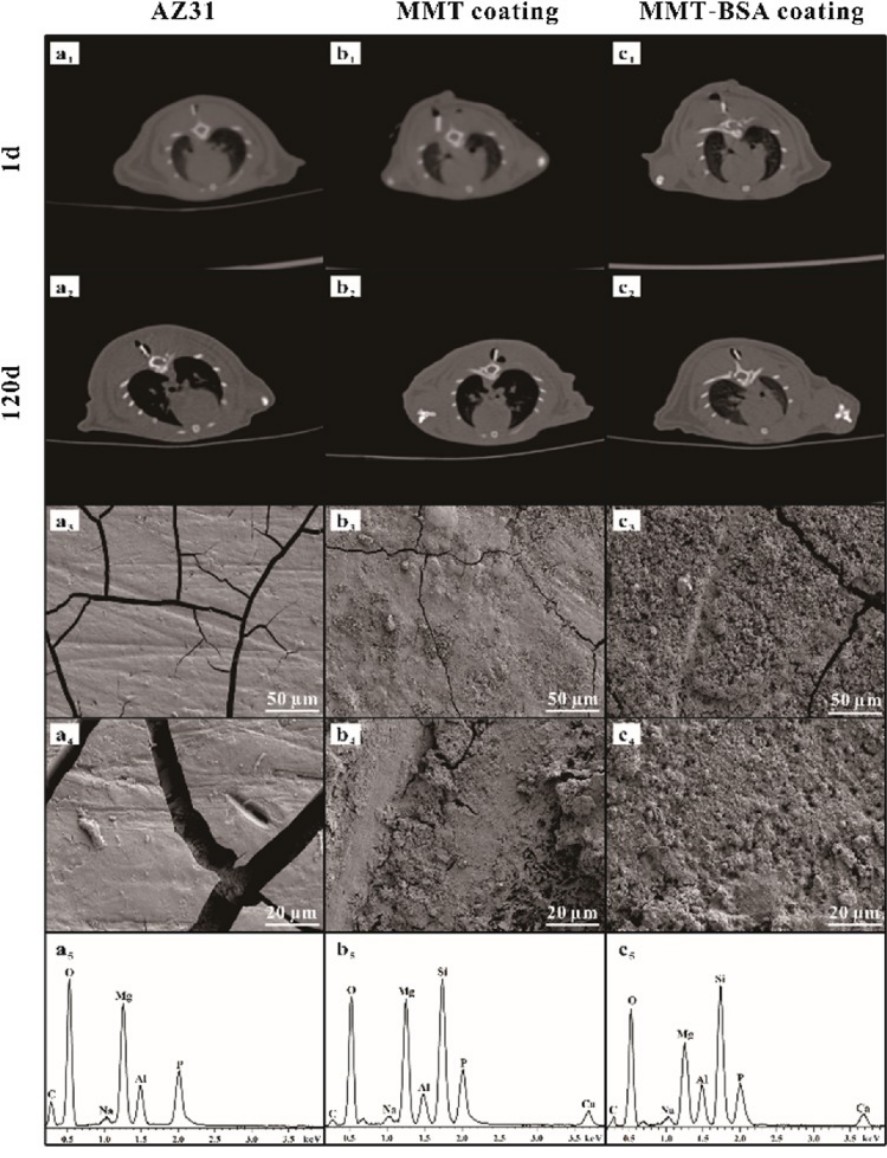

**Figure 18.** AZ31 with montmorillonite (MMT) coating and MMT–bovine-serum-albumin (BSA) coating in vivo: (**a1–c1**) spiral CT scan 1 day; (**a2–c2**) 120 days; (**a3–c3**, **a4–c4**) SEM images 120 days; and (**a5–c5**) energy dispersive spectroscopy (EDS), reprinted with permission from [74].

The results from one such set of testing [74] used three groups of rats (*n* = 5, per group) with implants of (i) magnesium, (ii) magnesium with sodium montmorillonite (MMT) coating, and (iii) magnesium with an MMT/bovine serum albumin (BSA) composite coating in the individual groups. A

scan was taken 24 h after the implant, and again 120 days after the implant, displayed in Figure 18 below (a1,a2,b1,b2,c1,c2). After the scan, the rats were sacrificed, and the implants were removed for further analysis on degradation levels in the sample itself. Histological analysis was performed on various body parts to determine the impact of the magnesium rod after 120 days of implantation. The figure from the energy dispersive spectroscopy (EDS) scan Figure 18a5,b5,c5 demonstrates the impact of the inserted pins on the animal's physiology. In Figure 18b5,c5, the presence of calcium in the scan indicates the coating layer on the tested pins is diffusing out into the bloodstream of the test subject. Under the SEM images, the degradation of the surface of the pins can be seen, where magnesium displays significant cracking and degradation, Mg–MMT and Mg–MMT/BSA show comparatively fewer cracks to indicate the coating is not degrading as significantly as the magnesium itself.

The surface observations of the MMT–BSA coated magnesium indicated that it could be a valued option to pursue, as the coating itself showed very little wear while the blood testing performed after 120 days with an EDS suggested that degradation of the magnesium was occurring inside of the body.

Xu et al. [72] used rat specimens as well, inserting a pin of Mg, Mg coated by steam oxidation (Mg–SO), or Mg coated with micro arc oxidation (MAO) into the femoral bone and examined by radiographic imaging (Figure 19) to determine the healing of the bone, gas bubble development, and degradation of the pin at 4, 8, and 12 weeks, with a secondary examination by micro CT (Figure 20) at the end of 12 weeks. In the radiographic observations, the Mg and Mg–SO groups developed gas bubbles around the implant at the 4- and 8-week marks. By 12-weeks, the gas had been almost completely absorbed in the Mg–SO group, while growing in the Mg group. In the magnesium coated with micro arc oxidation (Mg–MAO) group, however, gas formation was significantly lower in the 4-, 8-, and 12-week marks, showing signs of healing by week 8.

The uncoated magnesium sample was found to have degraded the most, with the least volume of new bone growth at the end of 12 weeks. The Mg–SO group showed slightly less degradation than the Mg group, but displayed a greater volume of bone growth, with the greatest volume of bone growth was found in the Mg–MAO group, though the implant had displayed the least amount of degradation. Table 3 summarizes the implant volume change vs. new bone growth volume for each coating.

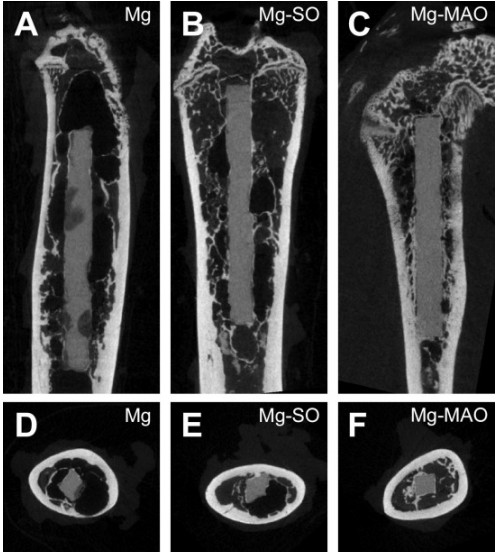

**Figure 19.** Micro-CT images of the rat femur implanted with the rod samples: A, B, and C are longitudinal views; D, E, and F are transverse views, reprinted with permission from [72]. MAO—micro arc oxidation.

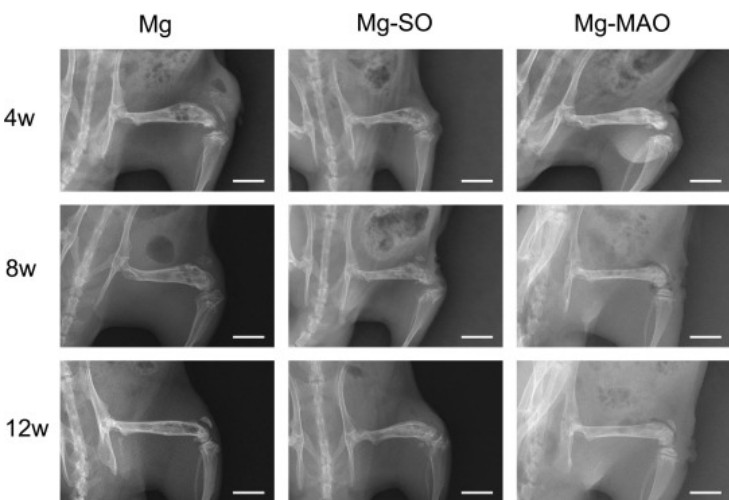

**Figure 20.** Radiographs of the Mg, Mg–SO, and Mg–MAO groups at 4, 8 and 12 weeks after the operation. Scale bar = 1 cm, reprinted with permission from [72].

**Table 3.** The change of new bone volume and the implant volume after indwelling for 12 weeks in rats.

| Coating Type | New Bone Volume (mm$^3$) | Initial Implant Volume (mm$^3$) | Final Implant Volume (mm$^3$) | Implant Volume Change (%) |
|---|---|---|---|---|
| Mg | 0.56 | 28.35 | 24.28 | −14.36% |
| Mg–SO | 1.52 | 28.35 | 26.32 | −7.17% |
| Mg–MAO | 4.72 | 28.35 | 27.68 | −2.38% |

An accepted downfall of animal testing is that it does not perfectly simulate the human body or the needs that arise for possible human subjects [63,84–87]. As a result, direct results for the human application can only be approximated until it is used in human clinical trials. In countries such as Germany, China, and South Korea, magnesium and magnesium alloy implants are being clinically tested to fix fractures and bone flaps, with Windhagen et al. [85] heading up the group with treatments already being used for hallux valgus surgery using screws made of a MgYReZr alloy [84,85]. Dewei et al. [86] performed testing with high purity magnesium screws in the hips of patients with stage II/III osteonecrosis in the femoral head (ONFH). During the 12-month follow-up period, the patients treated with the screws showed significant improvement over the group that had not been treated with the HP Mg screws [84,86].

Such in vivo trials can provide a qualitative comparison using CT or X-ray imaging to compare with models under development. Gartzke et al. [43] used in vivo μCT images for porous structures implanted in the femura of rabbits to compare qualitatively with their numerical model as shown in Figure 21.

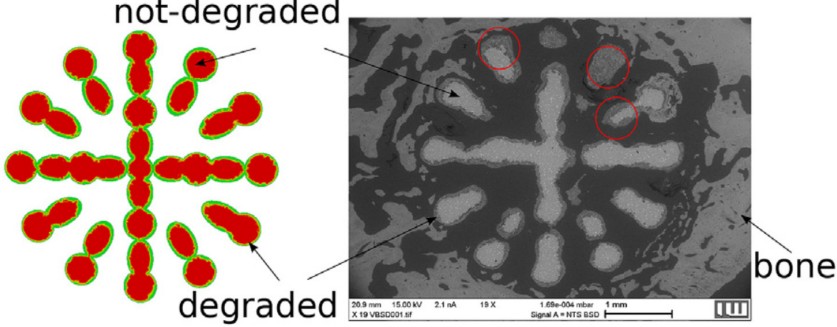

**Figure 21.** On the left is the numerical results showing magnesium concentration distribution after 8 weeks, and to the right the CT image of the implant after 6 weeks of implantation [43].

## 6. Conclusions

Magnesium and its alloys are the most investigated materials currently towards developing biodegradable implants. Part of the research effort in this field is focused on the development of an efficient method of testing these implants by using numerical modeling. Due to the complexity and the different mechanisms that are involved in the degradation process, a wide spectrum of hypotheses is currently being investigated in the literature to reach the closest match between the simulation results and the experimental in vivo and in vitro results. These simulation models are based on either phenomenological or physical modeling approaches. The phenomenological approach is based on the continuum damage theory and it provides an easier way to simulate the morphology change and the random pitting corrosion. On the other hand, the physical modeling approach is more capable of capturing the physical and electrochemical reactions with the environment occurring at the surface of the implants. Such reactions include the formation of precipitates on the surface, corrosion products, and coatings. Cellular automata (CA) represents a third potential modeling method that is not yet investigated for biodegradable materials modeling. Some examples of previous trials in literature using CA on different metals show promising modeling results almost matching random experimental data.

Though there are several experimental methods for testing the degradation behavior over time, some methods are more widely accepted than other methods. For instance, EIS and PDP are considered useful for approximating rates of degradation with the use of an electrical current. Both are fairly quick tests that can be performed in minutes, rather than the hours or days required for other methods of testing. The polarized methods are often used together in order to establish the corrosion rate expected from the alloy or pure magnesium in different environments. When using unpolarized methods, mass loss can be effective and most acceptable to provide the corrosion rate if the right procedure is followed per standards. Hydrogen evolution allows for testing to display how much magnesium is interacting with the corrosive fluid by collecting a volume of hydrogen gas, which can be measured and compared directly to the mass loss of the test sample, but will not account for magnesium lost from the sample through means other than corrosion with exposure to $H_2O$ molecules [88]. The use of pH monitoring has been established as ineffective when measuring the degradation rate of a sample of magnesium, as the position of the monitoring tool may not be fully indicative of the true pH balance of the corrosive material, depending on the placement of the measuring mechanism. In addition to this, pH is the main factor in the corrosion, so it needs to be adjusted during the test. Letting pH increase freely will affect simulation integrity.

Magnesium, magnesium alloys, and coated magnesium alloys have been used in in vivo testing with rats, rabbits, mice, and hamsters with promising results related to short-term and long-term degradation. By finding means of buffering the degradation of magnesium in an in vivo environment (either in animals or in human trials), safe healing tools can be established for use in human bone or cartilage regrowth treatments.

**Author Contributions:** Writing—original draft preparation, M.A. and A.J.; writing—review and editing, H.I.; visualization, H.I. and M.E.; supervision, H.I. and M.E.; funding acquisition, H.I. All authors have read and agreed to the published version of the manuscript.

**Funding:** This research received no external funding.

**Acknowledgments:** The authors acknowledge the support of the University of Tennessee at Chattanooga. Research reported in this publication was supported by the FY2020 Center of Excellence for Applied Computational Science competition.

**Conflicts of Interest:** The authors declare no conflict of interest. The funders had no role in the design of the study; in the collection, analyses, or interpretation of data; in the writing of the manuscript; or in the decision to publish the results.

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
