# Peer review of "Corrosion Modeling of Magnesium and Its Alloys for Biomedical Applications: Review"

_cmd, doi:10.3390/cmd1020011_

Round 1
Reviewer 1 Report
This is a comprehensive review of computational methods used for model Mg degradation used in biomedical field. The paper combines a good blend of background, model methods review, and experimental validation methods. Below are a few points the authors might consider.
- The authors try to categorize the models into phenomenological vs. physical. It is rather rare that one phenomenological model does not consider any physics/chemistry principle in the model construction. I suggest the authors reconsider better names to differentiate the two.
- For differentiating the different model types, I suggest adding a flow chart of model setup, input parameters, numerical methods, validation methods/parameters so it is easy to visualize the whole process, as well differences between the models.
- EIS method is used for coated Mg, as the author stated, it does not apply to uncoated Mg, which is the focus of this review.
- The in vivo section is loosely related to the theme, how is in vivo study used for validation of the models?
- For the experimental methods such as weight loss vs electrochemical potentiodynamic method, the authors clearly stated in the main text that potentiodynamic is not accurate for predicting corrosion rate, but the conclusion was written to imply that potentiodynamic method is better than weight loss.
- A brief discussion about model verification and uncertainty quantification could be added to strengthen the paper.
Reviewer 2 Report
I think the paper is a useful review. There are some errors in the introduction. The review of the models however is a good idea. I think others might find it beneficial.
Suggestions:
Since Fe and Zn are not reviewed, only Mg, then Fe and Zn should be removed. They are toxic in high concentrations, and not biocompatible.
[The number is the line number.]
48: United States is a proper noun, should be capitalized
49-50: $59.5 million seems low. Maybe billion instead.
51: Notation USD does not agree with $ in previous line
91: unfair comparing atmospheric corrosion to aqueous
119-120: Reducing the cathode size reduces corrosion.
151: I do not think Mg(OH)2 is on the bottom of the pit during corrrosion.
146: rupture, not rapture
181: Hard to believe this model is correct. The greater the corrosion damage, the higher the strength?
186: corrosion is a surface event, not throughout the depth.
665: Mg-SO not defined.
